# SASH: Efficient Secure Aggregation Based on SHPRG For Federated Learning

**Zizhen Liu**[1,3]      **Si Chen**[2]      **Jing Ye**[1,3]      **Junfeng Fan**[2]      **Huawei Li**[1,3]      **Xiaowei Li**[1,3]

[1]Institute of Computing Technology, Chinese Academy of Sciences, Beijing, China
[2]Open Security Research, Shenzhen, China
[3]CASTEST, Beijing, China

## Abstract

To prevent private training data leakage in Federated Learning systems, we propose a novel secure aggregation scheme based on seed homomorphic pseudo-random generator (SHPRG), named SASH. SASH leverages the homomorphic property of SHPRG to simplify the masking and demasking scheme, which for each of the clients and for the server, entails a overhead linear w.r.t model size and constant w.r.t number of clients. We prove that even against worst-case colluding adversaries, SASH preserves training data privacy, while being resilient to dropouts without extra overhead. We experimentally demonstrate SASH significantly improves the efficiency to 20× over baseline, especially in the more realistic case where the numbers of clients and model size become large, and a certain percentage of clients drop out from the system.

## 1 INTRODUCTION

In Federated Learning (FL), multiple participants collaborate to train a machine learning model without putting together their raw training data [Yang et al., 2019]. In the scenario of horizontal FL, a central coordinator updates the global model with the aggregation of the clients' local model updates. However, as recent studies argue, model inference attacks can compromise the privacy of training data from the information of model update [Zhu and Han, 2020, Hitaj et al., 2017], which puts forward requirements that the model update should be exchanged in a secure way. A secure aggregation solution for FL aims to solve this problem, and typically considers the following aspects:

1. Efficiency: the computation and communication cost introduced by the scheme, and the scalability to a large number of clients and parameters.

2. Security: the threat model of the scheme, including the goal of the adversaries, whether the adversaries collude, and the maximal number of colluding participants.

3. Practicality: the robustness of the scheme against client dropouts, and whether its implementation is compatible with a common Internet environment.

4. Accuracy: quality of the trained model, including peak accuracy and convergence speed.

Existing privacy solutions for FL apply privacy protection techniques, including Secure Multiparty Computation (SMC) [Bonawitz et al., 2017, So et al., 2020, Kadhe et al., 2020, Bell et al., 2020, Damgård et al., 2013, Truex et al., 2019], Homomorphic Encryption(HE) [Zhang et al., 2020, Kadhe et al., 2020], and Differential Privacy(DP) [Geyer et al., 2017]for various practical scenarios. However, when the number of clients, model size, and the dropout rate become large in real-world applications, it is still challenging to construct a secure and efficient aggregation scheme.

SecAgg [Bonawitz et al., 2017] is one of the most practical solutions to provide privacy guarantees in the horizontal FL. In SecAgg, for any pair of clients $u, v$ in the set of clients $\mathcal{U}$, they securely agree upon a masking seed $s_{u,v}$. Then for each client $u$ in $\mathcal{U}$, the seed of another mask is generated. The message of client $u$ is masked as $y_u = m_u + \mathrm{PRG}(b_u) + \sum_{v<u} \mathrm{PRG}(s_{u,v}) - \sum_{v>u} \mathrm{PRG}(s_{u,v})$. All the masking seeds $b_u$ and $s_{u,v}$ are also secret-shared among all clients, which allows the reconstruction upon sufficient shares. After receiving the masked values, the aggregator can reconstruct $b_u$ and remove $\mathrm{PRG}(b_u)$ from $y_u$ if $u$ is still online, or reconstruct $s_{u,v}$ for every other online client $v$ and remove all $\mathrm{PRG}(s_{u,v})$s from $y_u$ if $u$ drops out. The pairwise masking scheme entails computation complexities of $O(N^2M)$ for the aggregator, and $O(MN)$ for each client, where $N$ is the number of clients, and $M$ is the number of parameters in the model under training. This quadratic overhead in $N$ may limit its practical applications to FL systems with thousands of clients. Several subsequent works [So et al., 2020, Kadhe et al., 2020, Bell et al., 2020]

*Accepted for the 38th Conference on Uncertainty in Artificial Intelligence* (UAI 2022).

make the secure aggregation more efficient but will incur new restrictions, such as weakened dropout resilience or increased communication costs.

To further improve the efficiency of SecAgg, this work develops a novel secure aggregation scheme based on seed-homomorphic pseudo-random generator (SHPRG) [Chen et al., 2019], which has the property $\sum \text{SHPRG}(k_i) \approx \text{SHPRG}(\sum k_i)$. For each client, instead of masking the data with $N$ PRG outputs, one mask is sufficient. The masked data is $y_u = m_u + \text{SHPRG}(k_u)$, where $k_u$ is a self-generated seed. If the aggregator can securely get the sum of a subset of $\{k_u\}$, then the aggregator can remove the mask of the masked aggregation result by computing $\sum m_u = \sum y_u - \text{SHPRG}(\sum k_u)$. If some clients drop out during the process, the remaining clients' masked data are still valid, and the aggregator can get the correct sum, as long as the aggregator can securely get the sum of masking seeds of the surviving clients.

Overall, in this paper, we propose an efficient secure aggregation scheme for federated learning, named SASH. Our construction has the following traits:

- Masking and demasking are simpler and more efficient than the previous solutions, resulting in computation complexities of $O(M)$ for each of the clients and for the aggregator, and the communication cost is the same with SOTA schemes.

- Our scheme is robust to up to $D_{\max}$ arbitrary dropouts, and in the concrete instantiation, $D_{\max} = N/3$. In addition, the enhancement of efficiency is more significant when dropouts occur.

- The model aggregation is proved to achieve training data privacy against up to $T_{col}$ colluding clients, and in the concrete instantiation, $T_{col} = 2N/3$.

In the following sections, we give detailed construction of our secure aggregation scheme, and show that it achieves the four aspects of requirements for a practical and secure FL system.

## 2  PROBLEM STATEMENT AND BACKGROUND

In this section, we first formulate the problem we target, and then review previous research and the cryptographic primitives needed for our constructions.

### 2.1  PROBLEM STATEMENT

In this paper, we focus on the privacy of the typical horizontal federated learning, where $N$ data owners (also called clients) collaboratively train a model with $M$ parameters with the coordination of an aggregator (also called

the server). $N$ can range from a hundred to tens of millions [Kairouz et al., 2021] and $M$ may scale to millions [Bonawitz et al., 2019]. To avoid the inference of training data privacy from the exchanged model update $m_u$ during the learning process [Zhang et al., 2021], secure aggregation aims to learn $\sum m_u$ without revealing additional sensitive information beyond the model aggregation.

The threat model is honest-but-curious, and allows colluding. The potential adversaries in FL may be clients and the aggregator who can get access to the exchanged data. In colluding case, the adversary may control a set of up to $T_{col}$ clients, and may also control the aggregator. The independent or colluding adversaries can attempt to infer sensitive information based on the viewed intermediate data, such as the original individual model update, which can be utilized to infer the training data of some clients.

Dropout is another challenge that may interrupt secure aggregation. A random subset of up to $D$ clients may drop out of the system at any point of time during the execution of secure aggregation. It may fail model aggregation or result in a wrong global model.

In the proposed protocol, while keeping the trained model's accuracy unaffected, and keeping the implementation compatible with common Internet environment, we target to construct an efficient secure aggregation scheme, which protects the privacy of clients' data in colluding cases, and is robust again a significant portion of client dropouts.

### 2.2  RELATED WORK

We briefly review privacy solutions for horizontal FL in this section.

HE provides a general solution for security and privacy enhancements of FL [Zhang et al., 2020, Kadhe et al., 2020]. Many recent works advocate the use of additively HE schemes, notably Paillier [Paillier, 1999], as the primary means of privacy guarantee in FL. HE performs complex cryptographic operations that are relatively expensive to compute. The reference develops a simple batch encryption technique based on new quantization and encoding schemes to improve efficiency. However, questions arise about the collusion threats.

DP is a rigorous mathematical framework to improve the privacy of the machine learning model by introducing a level of uncertainty into the released model [Geyer et al., 2017]. With carefully added randomness to training data and/or trained models, DP can protect the privacy of individual samples in the dataset. DP can be used in combination with our scheme to provide further security guarantees.

SMC guarantees that a set of parties compute a function in a way that each one cannot learn anything except the output, and different SMC protocols such as SPDZ protocol

[Damgård et al., 2013] and threshold homomorphic encryption [Truex et al., 2019] have been utilized in the privacy-preserving FL framework. A notable work is the secure aggregation protocol proposed by Bonawitz et al. [Bonawitz et al., 2017]. As reviewed in section 1, they developed a double masking solution, which achieves secure aggregation against colluding participants, and is robust to dropouts. However, the quadratic growth of computation overhead w.r.t. $N$ is the major bottleneck. Several subsequent works improve the efficiency based on the framework of SecAgg. One-shot reconstruction of the aggregate-mask was employed in a recent work [Zhao and Sun, 2021], but can only work with a trusted third party(TTP). TurboAgg utilizes a circular communication topology to reduce the communication and computation overhead [So et al., 2020]. SecAgg+ achieves polylogarithmic communication and computation per client via communication graph [Bell et al., 2020]. FastSecAgg presents an FFT-based multi-secret sharing scheme to obtain $O(M \log N)$ cost [Kadhe et al., 2020]. However, in SecAgg+, TurboAgg, and FastSecAgg, the robustness to dropouts and/or security guarantees are weaker than those of the original SecAgg.

## 2.3 CRYPTOGRAPHIC TOOLS

**Seed Homomorphic Pseudorandom Generator.** Recall that a pseudorandom generator (PRG) is a deterministic polynomial-time algorithm $F : \{0,1\}^l \rightarrow \{0,1\}^n$ such that $l < n$, and for randomly distributed $s \in \{0,1\}^l$ and $r \in \{0,1\}^n$, the distributions of $F(s)$ and $r$ are computationally indistinguishable. A PRG $F : \chi \rightarrow \gamma$, where $(\chi, \oplus)$ and $(\gamma, \otimes)$ are groups, is said to be seed homomorphic if the following property hold [Boneh et al., 2013]: For every $s_1, s_2 \in \chi$, we have that $F(s_1) \otimes F(s_2) = F(s_1 \oplus s_2)$.

A Seed Homomorphic Pseudorandom Generator(SHPRG) can be constructed basing on the Learning With Rounding (LWR) problem as $G(s) = \lceil A^T \cdot s \rceil_p$, where $n, m, p, q$ satisfying $p < q, n < m$ are public parameters, $A$ is another public parameter randomly sampled from $\mathbb{Z}_q^{n \times m}$, and $\lceil \cdot \rceil_p$ is defined as $\lceil x \rceil_p = \lceil x \cdot p/q \rceil$ for $x \in \mathbb{Z}_q$. It is almost seed homomorphic in the following sense:

$$G(s_1 + s_2) = G(s_1) + G(s_2) + e, e \in [-1, 0, 1]^m. \quad (1)$$

Note that the security of the above SHPRG depends on the hardness of $\text{LWR}_{n,q,p}$ problem [Albrecht et al., 2015]. The value of $1/p$ is proportional to the error rate $\alpha$ in Learning With Error (LWE) [Regev, 2009], so the selection of parameters should assure that $\text{LWE}_{n,q,1/p}$ has difficulty satisfying the security level objective.

Multiple privacy-critical applications have been built from Seed Homomorphic PRG or the related preliminary Key Homomorphic Pseudorandom Functions, such as distributed PRFs, undatable encryption [Boneh et al., 2013] and private stream aggregation [Ernst and Koch, 2021, Valovich,

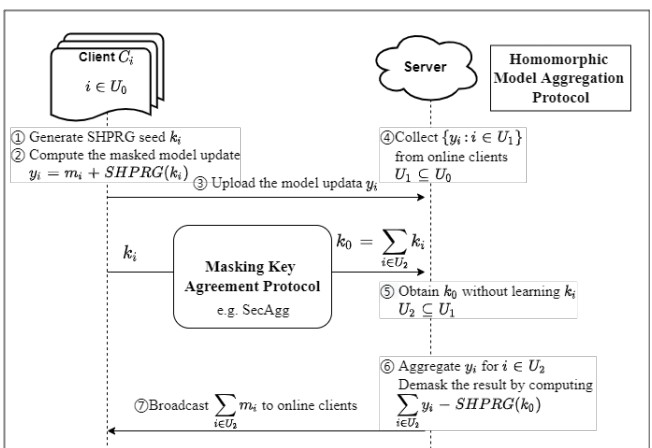

Figure 1: The Overall Process of SASH.

2017]. The homomorphism property is in support of specific applications with provable security.

## 3 SASH: SECURE AGGREGATION BASED ON SHPRG

In this section, we present an efficient privacy-preserving aggregation scheme based on SHPRG combining two layers of protocols: the Homomorphic Model Aggregation (HMA) protocol and the Masking Key Agreement (MKA) protocol. Figure 1 depicts the overall process of the mechanism for one epoch. The model updates are securely shared and computed following the HMA protocol, which calls the MKA protocol to return the demasking key to the aggregator and enable demasking to obtain the global model update. The process is repeated until the global model converges. Next, we will describe the two protocols in detail respectively.

### 3.1 THE HOMOMORPHIC MODEL AGGREGATION PROTOCOL

In the Homomorphic Model Aggregation Protocol, the aggregation of clients' local models is computed under the orchestration of the aggregator, ensuring no information about the individual models is revealed beyond their aggregated value. As shown in Figure 2, the inputs of HMA are model updates of all related clients in the initial set $\mathcal{U}_0$ which is demoted as $m_{\mathcal{U}_0}$. Each client $u$ firstly utilizes SHPRG to generate the mask $G(k_u)$ for the current epoch. They take the masking key, which is a randomly sampled vector, as the input to the SHPRG, and stretch it to a mask for each entry of the model update. Then they upload the masked model updates to the aggregator.

During this step for masking and uploading data, some clients may drop out. We denote the set of clients that have successfully uploaded masked data as $\mathcal{U}_1$. Clients in $\mathcal{U}_1$ and

---
**Homomorphic Model Aggregation Protocol**

**Parameter**: a random matrix $A \xleftarrow{R} \mathbb{Z}_q^{\mu \times M}, \mu, q, p, M \in \mathbb{N}$, with $q > p, \mu < M$

**Input**: $m_{\mathcal{U}_0} = \{m_u\}_{u \in \mathcal{U}_0}$ for the clients;

**Output**: $m_0 = \sum_{u \in \mathcal{U}_2} m_u$;

*Client $u$*:

1: Generate the masking key $k_u$ by sampling random vector of $\mu$ entries.

2: Preprocess the model update $m_u$ and encrypt the quantized model update $x_u$ to return $y_u = x_u + G(k_u) \mod P$.

3: Upload $y_u$ to the server.

*Server*:

1: Collect $y_u$ of all clients, and call the Masking Key Agreement protocol which returns $k_0 = \sum_{u \in \mathcal{U}_2} k_u$.

2: Do the aggregation $y_0 = \sum_{u \in \mathcal{U}_2} y_u$, and unmask it by computing $x_0 = y_0 - G(k_0)$.

3: Dequantize $x_0$ to obtain the final aggregation model update $m_0 = \sum_{u \in \mathcal{U}_2} m_u$, and broadcast the averaged aggregation $m_0/N_2$ to online clients for the next training.

---

Figure 2: The Homomorphic Model Aggregation Protocol.

the aggregator run the MKA protocol, and some further client dropouts may happen. We denote the set of surviving clients after MKA as $\mathcal{U}_2$, and the aggregator should obtain $k_0 = \sum_{u \in \mathcal{U}_2} k_u$ from the MKA protocol. The aggregator then sums up the model updates of clients in $\mathcal{U}_2$ and removes the mask which is $G(k_0)$. Subsequently, the aggregator dequantizes the result before computing the average over $N_2$ clients in set $\mathcal{U}_2$ and broadcasts the final aggregated model update.

We instantiate the protocol by the almost seed homomorphic PRG introduced in Section 2. Since the output of SHPRG is in $\mathbb{Z}_p^m$, we set the public modulus $P$ in our scheme equal to $p$. We quantize the model updates by converting each bounded local model update to $w$-bit integer before adding masks. For a model update $m$ in $[m_{\min}, m_{\max})$, the quantized value of $m$ is

$$Q(m) = \left\lfloor \frac{2^w (m - m_{\min})}{m_{\max} - m_{\min}} \right\rfloor. \quad (2)$$

where $\lfloor a \rfloor$ is the flooring function that maps $a \in \mathbb{R}$ to the largest integer not greater than $a$. The aggregation of quantized value over $N$ parties is at most $N(2^w - 1)$, so we set $p > N(2^w - 1)$ to make sure the summed model update does not overflow. For summation result $x$, the corresponding dequantization is performed by

$$Q^{-1}(x) = 2^{-w}(m_{\max} - m_{\min})x + N m_{\min}. \quad (3)$$

### 3.2 THE MASKING KEY AGREEMENT PROTOCOL

In the Masking Key Agreement (MKA) Protocol, each client $u$ holds the masking seed $k_u$, and the aggregator and online clients in $\mathcal{U}_2$ collaboratively compute the sum of the masking keys of online clients $\sum_{u \in \mathcal{U}_2} k_u$ without disclosing the

individual values to other clients or the aggregator. Various privacy protection approaches can be chosen to achieve secure aggregation.

In particular, the protocol of SecAgg [Bonawitz et al., 2017] can be called to implement MKA. The original SecAgg protocol performs secure aggregation of the model updates exchanged during the FL learning process, while as an implementation of MKA, it only aggregates the SHPRG seeds. SecAgg is robust against user dropouts less than some threshold, and the rest of HMA is not affected by user dropouts, so the whole protocol is tolerant to dropouts.

We can instantiate MKA with other secure aggregation solutions as well. As a tradeoff between efficiency and security/robustness, we can utilize SecAgg+ [Bell et al., 2020], FastSecAgg [Kadhe et al., 2020], or TurboAgg [So et al., 2020] to reduce the computation and communication overhead within MKA further. Note that in these schemes, the security and robustness are somehow weaker than those of SecAgg and the ideal HMA, and the chosen MKA scheme determines the security and robustness of the overall HMA scheme. If one wants to further guarantee privacy against malicious participants, this protocol can also combine authentication or correctness verification.

### 3.3 CORRECTNESS AND SECURITY

In this section, we state our correctness and security theorems. We consider clients in $\mathcal{U}_0$ and the sever $A$ execute the HMA protocol with inputs $m_{\mathcal{U}_0} = \{m_u\}_{u \in \mathcal{U}_0}$, $|\mathcal{U}_0| = N, \mathcal{U}_0 \supseteq \mathcal{U}_1 \supseteq \mathcal{U}_2$.

**Theorem 1 (Correctness)** *If participants in $\mathcal{U}_2$ follow the protocol, regardless of dropouts in $\mathcal{U}_0 \setminus \mathcal{U}_2$ (entries that are in $\mathcal{U}_0$ but not in $\mathcal{U}_2$), the server can obtain $\sum_{u \in \mathcal{U}_2} m_u$ with negligible noise based on the given $k_0 = \sum_{u \in \mathcal{U}_2} k_u$, where*

$|\mathcal{U}_2| = N_2$.

Proof: Because the selected PRG is almost seed-homomorphic, we have:

$$\sum\nolimits_{i=1}^{n} G(k_i) = G(\sum\nolimits_{i=1}^{n} k_i) + e \mod p$$
$$\text{where}, e \in \{-n+1, ..., 0, 1, ..., n-1\} \tag{4}$$

For the HMA protocol, $y_u = m_u + G(k_u) \mod P$, where $G(k_u) \in \mathbb{Z}_p^M, P = p, p \geq N(2^w - 1) + 1) \geq N_2(2^w - 1) + 1$, we have:

$$
\begin{aligned}
m_0 &= \sum_{u \in \mathcal{U}_2} y_u - G(k_0) \mod P \\
&= \sum_{u \in \mathcal{U}_2} (m_u + G(k_u)) - G(\sum_{u \in \mathcal{U}_2} k_u) \mod P \\
&= \sum_{u \in \mathcal{U}_2} m_u + \sum_{u \in \mathcal{U}_2} G(k_u) - G(\sum_{u \in \mathcal{U}_2} k_u) \mod P \\
&= \sum_{u \in \mathcal{U}_2} m_u + e_0 \mod P
\end{aligned}
\tag{5}
$$

where $e_0 \in \{-N_2 + 1, ..., 0, 1, ..., N_2 - 1\}$. The noise here is insignificant relative to the domain of aggregated quantized model updates ranging in $N_2(2^w - 1)$, which can be demonstrated to have a negligible impact on the quality of the trained model.

Theorem 2 below shows that HMA is secure against colluding participants, which may contain the aggregator, irrespective of how and when clients drop out. Those clients and the aggregator learn nothing more than their own inputs, and the sum of the inputs and masks of the other clients.

We consider executions of HMA with privacy threshold $T_{col}$, and underlying cryptographic primitives are instantiated with security parameters $\Lambda$. In such a protocol execution, the view of a client $u$ consists of its internal state (including its model update $m_u$, masking seed $k_u$, mask $G(k_u)$, the aggregated model update $\sum_{u'} m_{u'}$) and all messages this party received from other parties. The view of the server $A$ consists of the received information, including demasking seed $k_0$ and the masked model updates $\{y_u\}$ where $u \in \mathcal{U}_0$.

Given any subset $\mathcal{V} \subset \mathcal{U}_0 \cup A$, let $\mathsf{REAL}_{\mathcal{V}}^{\mathcal{U}_0, T_{col}, \Lambda}$ be a random variable representing the combined views of all parties in $\mathcal{V}$ in the execution of HMA, where the randomness is over the internal randomness of all parties, and the randomness in the setup phase. We show that for any such set $\mathcal{V}$ of honest-but-curious clients of size up to $N - 2$, the joint view of $\mathcal{V}$ can be simulated given the inputs of the clients in $\mathcal{V}$, and the sum of the inputs and masks of the other clients.

**Theorem 2 (Security)** *There exists a PPT simulator SIM such that for all $\mathcal{U}_0, m_{\mathcal{U}_0}, \mathcal{U}_1, \mathcal{U}_2$ and $\mathcal{V} \subset \mathcal{U}_0 \cup A, |\mathcal{V} \setminus \mathcal{A}| < N - 1$, the output of SIM is computationally indistinguishable from the joint view of $\mathsf{REAL}_{\mathcal{V}}^{\mathcal{U}_0, T_{col}, \Lambda}$ of the parties in*

$\mathcal{V}$:

$$
\begin{aligned}
&\mathsf{REAL}_{\mathcal{V}}^{\mathcal{U}_0, T_{col}, \Lambda}(m_{\mathcal{U}_0}, \mathcal{U}_1, \mathcal{U}_2) \approx \\
&\mathsf{SIM}_{\mathcal{V}}^{\mathcal{U}_0, T_{col}, \Lambda}(m_{\mathcal{V}}, z_m, z_k, \mathcal{U}_1, \mathcal{U}_2) \\
&z_m = \sum_{u \in \mathcal{U}_2 \setminus \mathcal{V}} m_u, z_k = \sum_{u \in \mathcal{U}_2 \setminus \mathcal{V}} G(k_u)
\end{aligned}
\tag{6}
$$

Proof: We prove the theorem by a standard hybrid argument. We will present a series of hybrids from variable REAL to SIM where any two subsequent random variables are computationally indistinguishable. We assume that $A \in \mathcal{V}$, which indicates the view of the server should be considered. The case of $A$ not in $\mathcal{V}$ is much easier to prove and is omitted for brevity.

$\mathsf{Hyb}_0$  In this hybrid, the variables are distributed exactly as in REAL. We choose a specific client $u'$ in $\mathcal{U}_2 \setminus \mathcal{V}$. For this client, based on the given $z_m$ and $z_k$, we can write as $y_{u'} = m_{u'} + G(k_{u'}) = z_m + z_k - \sum_{u \in \mathcal{U}_2 \setminus v \setminus \{u'\}} y_u$.

$\mathsf{Hyb}_1$  In this hybrid, for a party $u$ in $\mathcal{U}_2 \setminus \mathcal{V} \setminus \{u'\}$, in HMA protocol instead of sending $y_u = m_u + G(k_u)$, we send $y_u = m_u + P_u$, where $P_u$ is uniformly random. For $u'$, the masked data is still generated by $y_{u'} = z_m + z_k - \sum_{u \in \mathcal{U}_2 \setminus \mathcal{V} \setminus \{u'\}} y_u$. The security of SHPRG guarantees that the distribution of $\{y_u : u \in \mathcal{U}_2 \setminus \mathcal{V} \setminus \{u'\}$ is identically distributed to the corresponding one in $\mathsf{Hyb}_0$. On the other hand, $y_{u'}$ is determined by $\{y_u : u \in \mathcal{U}_2 \setminus \mathcal{V} \setminus \{u'\}\}, z_m$ and $z_k$, so the distribution of $\{y_u : u \in \mathcal{U}_2 \setminus \mathcal{V}\}$ is identically distributed to that in $\mathsf{Hyb}_0$.

$\mathsf{Hyb}_2$  In this hybrid, for party $u$ in $\mathcal{U}_0 \setminus \mathcal{U}_2 \setminus \mathcal{V}$, the simulator can just substitute their inputs of HMA protocol by uniform random vectors. Since the server will not do aggregation on their inputs and has no access to the values, the joint view of the parties in $\mathcal{V}$ does not depend on their inputs. Consequently, the joint view of the participants will be identical to the previous one.

$\mathsf{Hyb}_3$  In this hybrid, for party $u$ in $\mathcal{U}_2 \setminus \mathcal{V} \setminus \{u'\}$, we replace the uploaded data in HMA protocol by $y_u = P_u$, which is possible since $P_u$ was obtained in $\mathsf{Hyb}_3$ to be uniformly random, $m_u + P_u$ is also uniformly random. For the chosen client $u'$, its uploaded data is still computed by $y_{u'} = z_m + z_k - \sum_{u \in \mathcal{U}_2 \setminus \mathcal{V} \setminus \{u'\}} y_u$, which makes the joint view of clients in $\mathcal{U}_2 \setminus \mathcal{V}$ consistent with the previous one, and the joint distribution of the data uploaded by clients in $\mathcal{U}_2$ stays identical. Hence the joint view of the participants including the server is indistinguishable from the previous hybrid.

Thus, the PPT simulator SIM that samples from the distribution described in the last hybrid can output computationally indistinguishable from REAL, the distribution can be computed based on $m_{\mathcal{V}}, z_m, z_k$. The simulation does not restrict the number of joint viewed parties, which means HMA can

preserve the security against the aggregator colluding with an arbitrary subset of up to $N - 2$ clients.

# 4 EVALUATION

In this section, we perform a detailed evaluation of SASH from the perspectives of efficiency, accuracy, privacy security and practicality theoretically and experimentally. We compare SASH with SOTA methods [Bonawitz et al., 2017, Bell et al., 2020, So et al., 2020, Kadhe et al., 2020] in Table 1. We can see that SASH achieves the best asymptotic computation efficiency for both clients and the aggregator, and exceeds previous methods in other aspects as well.

Among the existing works, SecAgg is still the most practical method for achieving the best comprehensive performance, including security and robustness to dropouts. We select SecAgg as the baseline. SASH aims to overcome the efficiency bottleneck, without sacrificing other advantages.

## 4.1 EFFICIENCY

In this section, we analyze the computation and communication cost theoretically, and then conduct it by the experimental running time.

### 4.1.1 Analysis

**Computation Overhead**  In the HMA protocol, the computation cost is mainly derived from computing SHPRGs to generate masks for each entry in the model update vector. The computation costs for clients and the aggregator are both $O(M)$ regardless of the number of clients and the client dropout rate.

To analyze the efficiency improvement, we take SecAgg as an example of implementation. Recall that in SecAgg, the majority of computation cost comes from PRGs calculation expanding the various seeds to masks of $M$ entries. For each client, $N$ masks are required for one upload, entailing computation cost of $O(MN)$. For the aggregator, computation cost is $O(N^2M)$, which can be broken into $O(MN(1-d) + dMN^2(1-d))$, where $d$ is the fraction of dropped-out clients who present extra overhead for recovery, called as effective dropouts. Apparently, the computation cost of the aggregator increases quadratically with $N$ if $d$ is nonzero. The SecAgg in the MKA protocol is called to only aggregate the masking key of size $\mu$ instead of the model update with size $M$. The complexities of computation here are $O(N)$ and $O(N^2)$ for each client and the aggregator, respectively.

Our scheme presents a further improvement of efficiency in dealing with dropouts. For the comparison with SecAgg, in addition to the difference of computational cost w.r.t. $N$,

our protocol also incurs a much smaller effective dropout fraction $d_0$. This is because the masked model updates are uploaded before calling the MKA protocol, and the aggregator can aggregate and demask the model updates of remaining clients in $\mathcal{U}_2$ correctly without considering the dropping-out clients in $\mathcal{U}_0 \setminus \mathcal{U}_2$, while the dropout users in $\mathcal{U}_1 \setminus \mathcal{U}_2$ are handled by the SecAgg protocol. In other words, $d_0 \approx \frac{|\mathcal{U}_1 \setminus \mathcal{U}_2|}{|\mathcal{U}_0|}$. Since the clients may drop from the system at any time with a certain probability, the dropout fraction is positively correlated with the execution time of the corresponding process. The effective dropouts-related process in our solution is just the MKA protocol, while in SecAgg it includes the time-consuming step of masking and uploading model data. Therefore, the proportion of effective dropouts between our scheme and SecAgg is $\frac{d_0}{d} \approx \frac{C_N + \mu}{C_N + M}$, where $C_N$ is a constant related to the number of clients. For $N = 500, M = 100k$, we get $d_0 = d/7$ experimentally.

**Communication Overhead**  The main contribution to the communication traffic comes from the HMA protocol in which the communication cost is $O(M)$ for each client and $O(MN)$ for the aggregator, which is equal to the plain learning FL. The only communication involved is the upload of masked models of size $M \log_2 p$.

For the MKA protocol, the size of each masking key needed to be transferred securely is fixed to $\mu$. The communication cost differs with different secure aggregation solutions, all independent of the number of model parameters. If SecAgg is employed in the MKA protocol, four rounds of communication are needed in the MKA protocol, and the communication cost is $O(N)$ for clients and $O(N^2)$ for the aggregator. The total amount of transferred data of our scheme is dominated by the collection of masked model data, which is approximately $NM \log_2 p$. The inflation factor relative to the communication traffic of the plain FL learning system is $\frac{NM \log_2 p + T}{NMw} \approx \frac{\log_2 p}{w}$, where $T$ denotes the size of transferred data in the MKA protocol. For the selected protocol, the inflation factor is about 2.06 when $N = 500, M = 10^6, w = 16, p = 2^{32}$, which is the same as SecAgg. When $M$ or $N$ becomes larger, our inflation factor stays basically constant. While TurboAgg [So et al., 2020] requires at least $\log N$ rounds of communication, with notably increased communication overhead.

### 4.1.2 Experimental Results

To conduct the evaluation in experiments, we implement SASH and SecAgg in C++ with the following settings. We take $w = 16$, and set the parameter of SHPRG used in the HMA protocol as $\mu = 512, p = 2^{32}, q = 2^{64}$, for which the LWE evaluator estimates a hardness of over $2^{128}$. Also, $q/p \geq \sqrt{\mu}$, which ensures the LWR problem appears to be exponentially hard for any $p = \text{poly}(\lambda)$ as described by [Banerjee et al., 2012]. For the implementation of the MKA

Table 1: Comparison of Efficiency, Security and Dropout Guarantees of Our Proposed Scheme and the Related Works.

| | SecAgg | SecAgg+ | Turbo | FastSec | SASH |
|---|---|---|---|---|---|
| **Computation (Client)** | $O(MN + N^2)$ | $O(\log^2 N + M \log N)$ | $O(M \log N \log^2 \log N)$ | $O(M \log N)$ | $O(M + N^2)$ |
| **Computation (Aggregator)** | $O(MN^2)$ | $O(N \log^2 N + MN \log N)$ | $O(M \log N \log^2 \log N)$ | $O(M \log N)$ | $O(M + N^2)$ |
| **Communication (Client)** | $O(M + N)$ | $O(M + \log N)$ | $O(M \log N)$ | $O(M + N)$ | $O(M + N)$ |
| **Communication (Aggregator)** | $O(MN + N^2)$ | $O(MN + N \log N)$ | $O(MN \log N)$ | $O(MN + N^2)$ | $O(MN + N^2)$ |
| **Security** $T_{col}$ | Adaptive $2N/3$ | Non-adaptive parameter | Non-adaptive $N/2$ | Adaptive $N/10$ | Adaptive $N/3$ |
| **Dropout** $D_{\mathbf{max}}$ | Worst-case $N/3$ | Average-case parameter | Average-case $N/2$ | Average-case $N/10$ | Worst-case $N/3$ |

protocol, we choose the same cryptographic primitives as the original implementation in SecAgg. All experiments are run in a Lenovo server with the configuration of Ubuntu 20.04, Intel(R) Core i7-10700K 3.80GHz CPU and 32GB RAM.

We measure the running time of secure aggregation of a single FL epoch and compare SASH with the baseline. We further study the impact of model size, the number of clients, and dropout fraction. Since the secure aggregation is independent of the training process, synthesized vectors are used for locally trained models whose elements are encoded to 16-bit unsigned integers to test different model sizes better. The local training time is not included in the total running time, and the entire learning process can be deduced. We execute the tests 500 times and conclude the average running time and the standard deviation. As the results in Figure 3 illustrate, we conclude that:

1. SASH improves the efficiency, and more importantly, the running time of execution increases more gently when the number of both model parameters and clients increases. Hence, SASH can be scaled to the FL systems with millions of model parameters and thousands of clients.

2. The running time of SASH is relatively stable as the user dropout rate increases. The degree of the improvement significantly increases as the dropout fraction increases, and when $d = 0.3$, SASH provides a speedup of 20× over SecAgg. This gain is expected to increase further for larger $M$ and $N$.

SASH has better efficiency than the SecAgg, providing the same robustness and security. If we relax the requirements of security guarantee and robustness to dropouts, we can instantiate MKA with other efficient methods as well. We call our scheme that uses the SecAgg+ to implement MKA as the SASH+. SASH+ provides the same security and dropout resilience as SecAgg+, and improves the efficiency to much

more extent. The results of running time for different numbers of clients and different dropout rates are shown in Figure 4, which further demonstrates that SASH+ is more efficient than SecAgg+ in computation, especially when the number of clients and dropout rate become larger. As for the communication overheads, they are almost the same for k=100.

## 4.2 SECURITY

As proved in Section 3, SASH can provide privacy guarantees against the server colluding with an arbitrary subset of clients in the honest-but-curious setting. For the ideal aggregation scheme, if the aggregator corrupts a set of clients, the remaining clients' partial aggregation results will be disclosed. The information obtained by the colluding participants in SASH is the same as the ideal case.

The security of the HMA protocol in our scheme presents no restriction on the number of colluding parties, which means $T_{col}$ in our method depends on the implementation of the MKA protocol. As another aspect of privacy guarantee, SASH can mitigate adaptive adversaries. Recall that an adaptive adversary can choose the set of clients to corrupt during the protocol execution. In the proof of Theorem 2, the joint view of $\mathcal{V}$ can be simulated for any subset of parties without any restriction, so the adversary can adaptively choose the corrupted set at any stage of the protocol, which makes no difference to the conversion between hybrids and the final distribution of SIM. In comparison, in SecAgg+ and TurboAgg, since subsets of clients perform secure aggregation in stages, an adaptive adversary may corrupt all clients in such a subset, and cause information leakage. As summarized in Table 1, although SecAgg+ and FastSecAgg improve computation and communication efficiency to some extent, the security degrades.

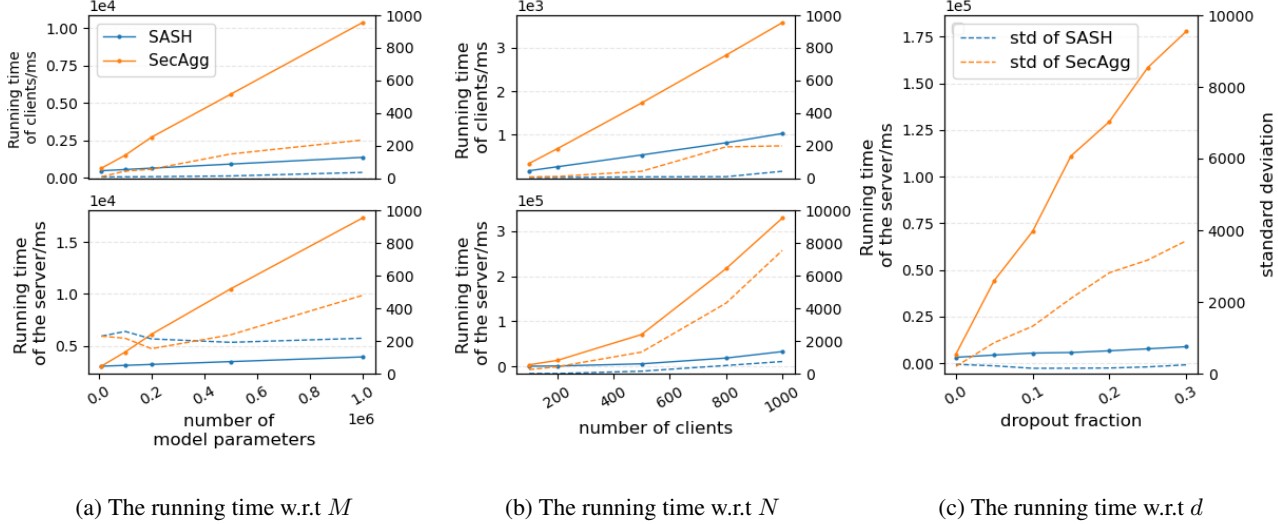

(a) The running time w.r.t $M$    (b) The running time w.r.t $N$    (c) The running time w.r.t $d$

Figure 3: Running Time of Executions. From left to right, (a) the running time as the number of model parameters increases for the FL system assembles 500 clients without dropouts. (b) the running time as the number of clients increases with M=100k and d=0.1. (c) the running time with dropout fraction with M=100K and N=50. The dotted lines represent the standard deviation of the results.

### 4.3 PRACTICALITY

As Table 1 illustrates, SASH can provide worst-case dropout resilience, which means the protocol can maintain correctness and security against any subset of up to $D_{max}$ clients dropping out. On the other hand, the average-case dropout robustness is limited to only random dropouts. For the defined dropout tolerance $D_{max}$, SecAgg+ [Bell et al., 2020] sets it as an adjustable variable, and larger $D_{max}$ demands more neighbors in the graph to provide security. In FastAgg [Kadhe et al., 2020], $D_{max} > N/10$ may result in failure to recover the secret, and $D_{max}$ must also be a constant fixed in advance. The proposed HMA protocol can be executed successfully with security guarantees for any $D < D_{max} = N - 1$, so the dropout tolerance is determined by the MKA protocol. Furthermore, as discussed in Section 4.1.1, MKA in our scheme incurs a much smaller effective dropout fraction, making the dropout condition of SASH $d_0 = d_0 N < D_{max}$ satisfied more easily.

Apart from the solid dropout guarantee, the execution of SASH does not assume of the existence of a Trusted Third Party (TTP). Also, there is no direct communication between clients, making our scheme easy to implement in real world.

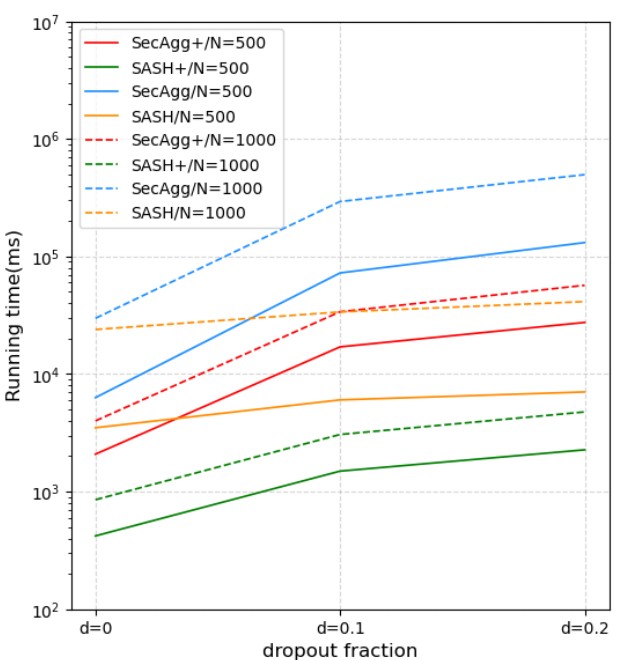

Figure 4: Comparison between SecAgg+, SASH+, SecAgg and SASH.

### 4.4 MODEL ACCURACY OF FL SYSTEM

We evaluate the impact of the noise produced by almost homomorphic PRG on the model accuracy of the FL system. We implement three representative machine learning

applications in FL, and perform plain aggregation and our proposed secure aggregation for each one. Our first application is a CNN model consisting of two convolutional layers with a total of about 0.2M parameters, trained over the FashionMNIST dataset [Xiao et al., 2017]. In another application, we train ResNet18 [He et al., 2015] with 10M parameters on the CIFAR10 dataset [Krizhevsky et al., 2009]. In the third application, we use Shakespeare dataset [Caldas et al., 2018] to train a customized LSTM [Hochreiter and Schmidhuber, 1997] with 1.25M parameters. The three applications are based on different types of machine learning models of various sizes, and cover the learning task for image classification and text generation. The optimization approach for federated learning is the Federated Averaging algorithm [McMahan et al., 2017]. For plain FedAvg aggregation, the model updates are represented by real-valued vectors of 32 bits and uploaded for aggregation without encryption.

For SASH, the model updates have two sources of error: (1) the model parameters are quantized into 16-bit integers before masking, and corresponding dequantization is done after aggregation; (2) SHPRG induces an error term to aggregated model parameters. To measure the model quality, we track the test accuracy for CNN and ResNet18. Training loss is used for LSTM as the dataset is unlabelled and has no test set. As Figure 5 shows, for one thing, compared with plain FedAvg, the convergence achieves after training for the same epochs, which means the speed of convergence is not affected. For another thing, the trained models obtained by SASH reach the same peak accuracy or bottom loss with the plain FedAvg.

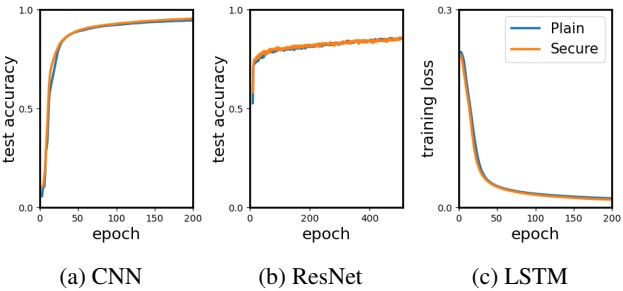

(a) CNN     (b) ResNet     (c) LSTM

Figure 5: The Quality of Trained Model

## 5 CONCLUSION

This paper presents an efficient and practical secure aggregation scheme based on SHPRG. We demonstrate our scheme from the following aspects: (1) our scheme achieves better asymptotic computation costs than previous solutions, and improves the efficiency up to 20× over baseline experimentally. (2) the proposed scheme is proved to provide adaptive security against the aggregator colluding with an arbitrary subset of clients. (3) our scheme is robust to worst-case

dropouts and simple to implement in a standard Internet environment for non-TTP assumptions. (4) the trained model can obtain the same accuracy as plain training cases.

For future work, an extension of the scheme to cross-silo FL settings, and the Byzantine-robustness of the scheme can be investigated.

## Acknowledgements

This paper is supported in part by the National Key Research and Development Program of China under grant No. 2020YFB1600201, National Natural Science Foundation of China (NSFC) under grant No. (U20A20202, 62090024, 61876173), and Youth Innovation Promotion Association CAS.

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
