# OpenReview forum: "SASH: Efficient Secure Aggregation Based on SHPRG For Federated Learning"
_auai.org/UAI/2022/Conference — UAI 2022 Poster_

### Official Review · Reviewer_XJC6 · 2022-04-14

**Q2(1) Originality/Novelty:** 3
**Q2(2) Significance/Impact:** 2
**Q2(3) Correctness/Technical Quality:** 2
**Q2(6) Clarity Of Writing:** 2
**Q6 Overall Score:** 6
**Q8 Confidence In Your Score:** 3

**Q1 Summary And Contributions:**

in this paper, the authors propose an aggregation scheme for federated learning, named SASH to prevent private training data leakage in Federated Learning systems.

**Q2 Assessment Of The Paper:**

More detailed information regarding each of these aspects is given below:

**Q2(4) Quality Of Experiments (Optional):**

2: Fair: The experimental evaluation is weak: important baselines are missing, or the results do not adequately support the main claims.

**Q2(5) Reproducibility:**

2: Fair: Key resources (e.g., proofs, code, data) are unavailable but key details (e.g., proof sketches, experimental setup) are sufficiently well-described for an expert to confidently reproduce the main results.

**Q3 Main Strengths:**

This paper presents an aggregation scheme based on SHPRGs and according to the authors, the proposed scheme (1) achieves better asymptotic computation costs than previous solutions, (2) provides adaptive security against the aggregator, (3) is robust to worst-case dropouts and simple to implement in a standard Internet environment for non-TTP assumptions and (4) the trained model can obtain the same accuracy as plain training cases.

**Q4 Main Weakness:**

Lack of statistical analysis and sharing a Git repository with the code to easily replicate the experiments.

**Q5 Detailed Comments To The Authors:**

Fig. 1 is somehow confusing and it is not properly explained in thee body of the manuscript.
Table 1, exceeds the limits of the page.
In Fig, 3,4 could use subcaptions.

**Q7 Justification For Your Score:**

Authors selected SecAgg as the baseline and SASH aims to overcome the efficiency bottleneck, without sacrificing other advantages.

**Q9 Complying With Reviewing Instructions:**

1: Yes.

---

### Official Review · Reviewer_sVWk · 2022-04-15

**Q2(1) Originality/Novelty:** 3
**Q2(2) Significance/Impact:** 3
**Q2(3) Correctness/Technical Quality:** 3
**Q2(6) Clarity Of Writing:** 3
**Q6 Overall Score:** 5
**Q8 Confidence In Your Score:** 2

**Q1 Summary And Contributions:**

This paper studies the problem of secure aggregation in federated learning. The main contribution of the current paper is a more efficient method.

**Q2 Assessment Of The Paper:**

More detailed information regarding each of these aspects is given below:

**Q2(4) Quality Of Experiments (Optional):**

2: Fair: The experimental evaluation is weak: important baselines are missing, or the results do not adequately support the main claims.

**Q2(5) Reproducibility:**

2: Fair: Key resources (e.g., proofs, code, data) are unavailable but key details (e.g., proof sketches, experimental setup) are sufficiently well-described for an expert to confidently reproduce the main results.

**Q3 Main Strengths:**

A more efficient secure aggregation method for federated learning has been established.

**Q4 Main Weakness:**

The experiments lack several baselines.

**Q5 Detailed Comments To The Authors:**

The problem considered in this paper is very important, and the proposed more efficient method looks very promising. However, I have one question: Why do authors not compare with those baselines in Table 1?

**Q7 Justification For Your Score:**

The proposed method seems to be much more efficient than existing method.

**Q9 Complying With Reviewing Instructions:**

1: Yes.

---

### Official Review · Reviewer_L13t · 2022-04-18

**Q2(1) Originality/Novelty:** 2
**Q2(2) Significance/Impact:** 2
**Q2(3) Correctness/Technical Quality:** 3
**Q2(6) Clarity Of Writing:** 2
**Q6 Overall Score:** 5
**Q8 Confidence In Your Score:** 3

**Q1 Summary And Contributions:**

This work proposes a novel secure aggregation scheme for horizontal FL. It has a computational complexity of O(M) for each client and allows D_max=N/3 arbitrary dropouts or T_col=N/3 colluding clients.


**Q2 Assessment Of The Paper:**

More detailed information regarding each of these aspects is given below:

**Q2(4) Quality Of Experiments (Optional):**

2: Fair: The experimental evaluation is weak: important baselines are missing, or the results do not adequately support the main claims.

**Q2(5) Reproducibility:**

2: Fair: Key resources (e.g., proofs, code, data) are unavailable but key details (e.g., proof sketches, experimental setup) are sufficiently well-described for an expert to confidently reproduce the main results.

**Q3 Main Strengths:**

1. Solid theoretical results which justify the efficiency and effectiveness of the proposed method SASH.
2. Experiments show SASH has significant improvement over SecAgg in terms of efficiency while maintaining very similar utility.

**Q4 Main Weakness:**

1. Only one baseline is compared in the experiments although several are listed in Table 1.
2. Only one set of parameters of SHPRG is considered in the experiments.

**Q5 Detailed Comments To The Authors:**

1. eq.(6) is out of bound.
2. Table 1 does not fit the template, too.
3. Fonts in Fig. 4 are too small.
4. It would be better to show the confidence interval/standard deviation of the results in Fig. 3 and 4.

**Q7 Justification For Your Score:**

1. Theoretical results look solid.
2. Experimental results are satisfactory but it needs further comparison with different baselines and different settings.

**Q9 Complying With Reviewing Instructions:**

1: Yes.

---

### Decision · Program_Chairs · 2022-05-15

**Decision:**

Accept (Poster)

**Comment:**

Meta Review: The paper proposes a secure aggregation scheme to prevent private data leakage in Federated Learning systems.

Quality: The technical discussions are sound. Experiment evaluation is weak.

Clarity: The paper is well written with some minor concerns on clarity.

Originality:  A novel application in Federated Learning.

Significance: The work is useful in Federated Learning.

A common concern is that the experiments have been conducted in comparison to one baseline and using one set of parameters. The authors have updated the experiments in their replies.